# A Recommendation Model Based on Multi-Emotion Similarity in the Social Networks

**Jun Long** [1] , **Yulou Wang** [1], **Xinpan Yuan** [2,*], **Ting Li** [1] **and Qunfeng Liu** [1]

1    School of Information Science and Engineering, Central South University, Changsha 410083, China;
     jlong@csu.edu.cn (J.L.); 174612284@csu.edu.cn (Y.W.); tinglicsu@csu.edu.cn (T.L.);
     qunfengliu@csu.edu.cn (Q.L.)
2    School of Computer, Hunan University of Technology, Zhuzhou 412000, China
*    Correspondence: xpyuanfly@163.com; Tel.: +86-0731-8253-9926

**Abstract:** This paper proposed a recommendation model called RM-SA, which is based on multi-emotional analysis in networks. In the RM-MES scheme, the recommendation values of goods are primarily derived from the probabilities calculated by a similar existing recommendation system during the initiation stage of the recommendation system. First, the behaviors of those users can be divided into three aspects, including browsing goods, buying goods only, and purchasing–evaluating goods. Then, the characteristics of goods and the emotional information of user are considered to determine similarities between users and stores. We chose the most similar shop as the reference existing shop in the experiment. Then, the recommendation probability matrix of both the existing store and the new store is computed based on the similarities between users and target user, who are randomly selected. Finally, we used co-purchasing metadata from Amazon and a certain kind of comments to verify the effectiveness and performance of the RM-MES scheme proposed in this paper through comprehensive experiments. The final results showed that the precision, recall, and $F_{1\text{-measure}}$ were increased by 19.07%, 20.73% and 21.02% respectively.

**Keywords:** recommendation model; social network; multi-emotion; cold start

---

## 1. Introduction

In recent years, people have been doing more online shopping on sites such as Amazon, Taobao, and Jingdong. As a result, how to build up an effective recommendation model is becoming a crucial research project [1]. Recommendation systems in social networks were first proposed by Resnick P and Varian HR in 1997 and used to provide personalized and intelligent information services to users on online shopping sites. A variety of recommendation systems have been proposed by researchers. The main recommendation systems include [2–5]: (1) Content-based recommendation systems, which will recommend the goods that a user is interested in based on their historical behaviors; (2) collaborative filtering recommendation systems, which adopt the similarities of users' historical purchasing behaviors to better represent the recommendation process in social networks; (3) hybrid recommendation systems. The online shopping website often takes advantage of multiple methods to improve its recommendation ability.

To the shop owner, it is important that a recommendation system can effectively introduce products with potential purchasing power to users. Although there are various methods which have been recommended in previous studies, some significant further difficulties are still needed to be overcome. For example, the "cold start" problem still exists in recommendation schemes and is not easy to be solved effectively. When a new shop opens, although it does not have purchase records, the relationships among goods are established by referring to identical products of the

selected existing shop. The existing store is one that has been running for some time and whose historical purchase records are rich. For newly opened stores with nonexistent or sparse transaction records, it is difficult for the recommendation system to make an effective recommendation. Therefore, the existing store that is most similar to the target store is chosen as the reference existing store in the paper. The calculation of similarity is the number of goods in the new store divided by that in the existing store. The reference existing store shares the maximum numbers of goods with the target store; therefore, we can recommend goods to target users according to the reference existing store. Emotion, as an indispensable psychological activity in social networks, always affects the daily lives and decision-making processes of users in shopping. This paper proposes a recommendation model (RM-MES) based on multi-emotion similarity in social networks. The problem studied in this paper targets a particular store and regards how to effectively recommend goods to users to maximize the benefits of the shop owner and how to improve the performance indexes of the recommended scheme, such as the precision, recall, and $F_{1\text{-measure}}$.

Our main contributions of this scheme are as follows:

(1) To solve the "cold start" problem, the RM-MES scheme uses the historical purchase records of an existing store to guide a recently opened store, which aims to form a recommendation probability matrix of both the existing store and the new store for the target users;

(2) To improve the accuracy of recommendation results, we propose a scheme based on multi-emotional analysis. The LDA topic model is used to subdivide user evaluation into six indexes. Considering user preferences for different levels of goods, the similarity of users is deeply analyzed, and the similarity results show its advantages;

(3) With the considerations of the different performances of users, the behaviors of those users can be divided into three aspects, including browsing goods, buying goods only, and purchasing–evaluating goods. According to the three categories, the browsing similarity, purchasing similarity, and emotional similarity among users can be identified;

(4) We adopt the metadata of Amazon goods to verify the effectiveness and performance of the RM-MES scheme through comprehensive experiments. In addition, we analyze the impact of transition probability influence factor $\alpha$ through the experiments.

## 2. Related Works

Generally, recommendation systems use a certain algorithm based on user behavior data or item data to recommend items that users need. According to the differences of the recommendation algorithms, the recommendation systems can be divided into the following categories [6–11]:

(1) Content-based recommendation systems. According to the items that users have liked, a content-based system can recommend similar items to users. Such systems were developed based on information retrieval and filtering, using the historical purchase records of target users or analysis of the characteristics from purchase information via statistics and machine learning. Chen et al. [12] proposed a probabilistic approach on the basis of TrueSkill for content-based recommendation systems. This system is useful for handling high uncertainty because it is only based on available goods and ratings given by users. There are still some disadvantages, such as the limited content analysis and the new user problem.

(2) Collaborative filtering recommendation systems, which are one of the most widely-used methods in practical applications, and their practical applications include Amazon, Taobao, and Digg. These types of schemes recommend products based on other users that have relationships that are similar to those of the target user. Li et al. [13] designed a trust-aware recommender system, which fully extracted the influence of trust information and contextual information on ratings to improve precision. Wang et al. [14] designed a combination model composed of the recommender and the similarity measure. He et al. [15] proposed a novel model for the one-class collaborative filtering setting, which combines high-level visual features extracted from a deep convolutional neural network, users' past feedback, as well as evolving trends within the community to uncover the complex and

evolving visual factors that people consider when evaluating products. Sun et al. [16] proposed a time-sensitive collaborative filtering method to discover the latest preferences of the customers and improve the accuracy of the recommendation system without complicating the training phase. As a typical recommendation method, collaborative filtering recommendation systems still have some problems that need to be addressed, such as the sparse database problem and "cold start" problem.

(3) Hybrid recommendation systems, which combine the advantages of each recommendation scheme. As each recommendation scheme is not perfect, hybrid recommendation systems are frequently used in practical applications. Not all combination methods are effective in practical applications. It is important to avoid or compensate for the weaknesses of their recommendation. In the combination method of hybrid recommendation systems, researchers have proposed seven ideas of combination: Weight, switch, mixed, feature combination, cascade, feature augmentation, and meta-level. Song et al. [17] researched how to gain better recommendations of traditional recommendation models on the basis of relationship information in social networks between customers and shops and proposed a matrix decomposition framework based on integrating relationship information in social networks.

Emotion, as an indispensable psychological activity in social networks, always affects the daily lives and decision-making processes of users. Recommendations based on emotion get much attention from researchers in the field of personalized recommendation [18–22]. Guo-Qiang et al. [23] built a collaborative filtering recommendation algorithm based on user emotion and combined user ratings and emotional comments together through subject extraction and sentiment analysis in users' project reviews. Wijayanti et al. [24] proposed an ensemble of a machine learning approach to detect the sentiment polarity in user-generated text. Vagliano et al. [25] proposed a recommendation method according to the semantic annotation of entities that are recorded in customer comments, and the entities are considered as candidate recommendations. Musto et al. [26] designed a multi-criteria collaborative filtering method, which uses aspect-based sentiment analyses of users' reviews to obtain sentiment scores as ratings of items from users. Contratres et al. [27] proposed a recommendation process that includes sentiment analysis to textual data extracted from Facebook and Twitter and presented the results of an experiment in which this algorithm was used to reduce the cold start issue. Seo et al. [28] proposed a friendship strength-based personalized recommender system. The personalized recommender system grants a weight to those users who are closely connected in their social circle based on friendship strength in order to recommend the topics or activities in which users might be interested. Meng et al. [29] provided a principled and mathematical way to exploit both positive and negative emotion on reviews and proposed a novel framework MIRROR, exploiting emotion on reviews for recommend systems from both global and local perspectives.

These schemes above have further improved the effectiveness of recommendation algorithms. However, these recommendation methods still have some problems that need to be overcome:

1.  Most of the recommendation schemes only consider the "cold start" problem of new users, but do not consider the "cold start" problem for a recently opened store, so as to affect the recommend quality of recommendation system;
2.  Some recommendation schemes search for user preferences by extracting user Facebook and Twitter data. However, it is difficult to extract the user's personal information due to issues such as permissions and technology. Additionally, because information that includes user emotions is often incomplete and fuzzy, it is not easy to directly analyze the emotions in the information from Facebook and Twitter;
3.  These recommendation systems based on emotion only consider positive and negative emotions but do not consider users' preferences in other aspects;
4.  When calculating the similarities of users' behaviors, most recommendation schemes do not take the correlation between projects into consideration;
5.  Most recommendation schemes fail to consider the trust factor of each piece of merchandise, which may cause the recommendation system to provide distrusted items to target users.

## 3. The RM-MES Algorithm

In the RM-MES scheme, the set of users is defined as $C = (c_1, c_2, \ldots, c_n)$, the set of shops is defined as $S = (s_1, s_2, \ldots, s_n)$, the set of goods in the new target shop is defined as $I = (I_1, I_2, \ldots, I_n)$, the set of goods in the reference existing shop is defined as $ref = (r_1, r_2, \ldots, r_n)$, and the set of reviews of users is defined as $C_i = (c_{i1}, c_{i2}, \ldots, c_{in})$.

In this paper, the relevant notations are shown in the Table 1.

**Table 1.** Main notations.

| Symbol | Description |
|---|---|
| $Sim_{c[a]}$ | The set of similar users to the target user $a$ |
| $SMX_{I[a]}$ | The purchase matrixes of similar users |
| $\lambda_{i,j}^{S}$ | The relationship among good $i$ and good $j$ |
| $\cdot$ | The proportion of the mean recommendation probability |
| $n$ | The number of final purchases in the new shop |
| $List_i$ | The number of recommended goods in each round |
| $w$ | The length of the time window |
| $y$ | The proportion of the influence factor of trust |
| $z$ | The proportion of the influence factor of the latent factor |
| $H[i][j]$ | The recommendation matrix of the target user based on the similarity of users |
| $S[i][j]$ | The recommendation matrix of the target user based on the correlation relationship among goods |
| $A[i][j]$ | The recommendation probability matrix of the target user based on $M[i][j]$ and $S[i][j]$ |
| $trust[i]$ | The value of trust for good $i$ |
| $rep[i]$ | The reputation of good $i$ |
| $fre[i]$ | The purchase frequency of good $i$ |
| $\mu$ | The proportion of the recommendation probability for the new shop |
| $recall$ | The probability that users purchase what they like in the recommendation list |
| $F_{1-measure}$ | The standard measurement for the classification accuracy of a recommendation algorithm |
| $B_i$ | The number of goods that user $i$ likes |
| $N_i$ | The number of goods that user $i$ has purchased in the recommendation list |

*3.1. Search for Existing and Similar Reference Users in the Existing Shop for the Taget User*

### 3.1.1. The Calculation Method for Similar Shops

First, the RM-MES scheme should search the existing store that is most similar to the target store for reference, which means finding the largest number of goods that the target store and existing store both have in common. These typical stores can be calculated with Equation (1) below:

$$R[i] = \frac{num(S) \cap num(S_i)}{num(S) \cup num(S_i)},\qquad(1)$$

where $num(S_i)$ represents the number of goods that existing shop $S_i$ has and $num(S)$ represents the number of goods that the new target store has. In the experiment, we chose the most similar store $S_i$ for reference, which means searching the maximum result of $R[i]$ according to Equation (1).

### 3.1.2. Emotional Analysis of User Reviews

The first step is data preprocessing. The reviews of user are first categorized on the basis of their attributes. Latent Dirichlet allocation (LDA) [20] has been employed as a technique to identify and annotate large text corpora with concepts, to track changes in topics over time, and to assess the similarity between documents. The LDA topic models provide the identification of core topics from a provided text collection. By analyzing the LDA thematic model of 5000 online reviews, we found that most consumers pay attention to six indicators: Quality, price, appearance, configuration, service, and express delivery. We thus classified reviews of users into six respective categories.

The second step is to extract emotional information from the reviews of users. This includes the extraction and discrimination of evaluation words, the extraction of evaluation objects, the extraction of combination evaluation units, the extraction of evaluation phrases, and the extraction of evaluation collocations. Then, based on an emotional lexicon, we analyzed user emotional polarity and obtained emotional values.

To distinguish words with the same emotional tendencies and different emotional polarity, we obtained the emotional scores of emotional words according to the public emotional vocabulary of HowNet (http://www.keenage.com/download/sentiment.rar). The HowNet is popular due to its context-specific lexicons. There are three categories of words: Emotional words, degree words, and negative words. Negative words can be used to determine whether the polarity of a comment is reversed or not. Degree words can provide different scores to different emotional words, and emotional words can be divided into positive words and negative words. If an emotional word is not in HowNet or has no emotional value, then we found its synonyms on the basis of TongYiCi Cilin (Mei et al., proposed in 1983) and compute the relevant emotional score. The text grading formula, as shown in Equation (2):

$$Score(i) = \sum_{j}^{n} (-1)^t * k * word(j) \tag{2}$$

where $Score(i)$ represents the score of each comment, the index $t$ of $-1$ depends on polarity reversal, $k$ represents the degree of degree word, and $word(j)$ is the original score of every word.

Finally, we computed the value of the comment:

$$Rep_i = \omega_1 Score(quality) + \omega_2 Score(price) + \omega_3 Score(appearance)$$
$$+ \omega_4 Score(configuration) + \omega_5 Score(service) + \omega_6 Score(delivery), \tag{3}$$

where $Rep_i$ represents the reputation of the commodity, $Score$ represents the emotional score of the evaluation, and $\omega_i$ represents the weight of each index.

### 3.1.3. The Calculation Method for Similar Users

According to the flow of information in social networks, the target user is randomly selected by the RM-MES scheme, and then we need to find users that are similar to the target user. When considering the similarity of user behavior, most schemes ignore the different performance of users; therefore, the precision of recommendation results may not be satisfactory. After taking into account the different performances of users, we divided users' behaviors into browsing goods, buying goods, and purchasing goods as well as evaluating these goods. Then, we obtained the similarity of their browsing and purchasing behaviors and emotional feelings among two users.

To obtain the similarity between $c_a$ and target user $c_b$, we first obtained the similarity of their browsing. The browsing similarity formulas are as follows:

$$Sim1_{a,b} = Sim(Browse(c_a), Browse(c_b)) = \frac{Browse(c_a) \cap Browse(c_b)}{Browse(c_a) \cup Browse(c_b)}, \tag{4}$$

where $Browse(c_a)$ represents the goods that the user $c_a$ has browsed and $Browse(c_b)$ represents the goods that the user $c_b$ has browsed. To obtain the similarity of their purchasing between user $c_a$ and target user $c_b$, Equation (5) can be obtained as follows:

$$Sim2_{a,b} = \frac{\sum_{k \in S_{a,b}} \sqrt{\frac{1}{fre_k^2}} (r_{a,k} - \overline{r_i})(r_{b,k} - \overline{r_j})}{\varphi_a \varphi_b}. \tag{5}$$

The similarity of emotional feelings between user $c_i$ and target user $c_j$, Equation (6) can be obtained as follows:

$$Sim3_{a,b} = \frac{\sum_{k \in S_{a,b}} \sqrt{\lambda \frac{1}{rep_k^2} + (1-\lambda)\frac{1}{fre_k^2}} (r_{a,k} - \overline{r_i})(r_{b,k} - \overline{r_j})}{\varphi_a \varphi_b}, \tag{6}$$

where $Simi_{a,b}$ represents the correlation of emotional similarity between user $c_a$ and user $c_b$, and $rep_k$ and $fre_k$ respectively represent the reputation and frequency of good $I_k$. $S_{a,b}$ represents the set of goods that were purchased both by user $c_a$ and user $c_b$. $\overline{r_i}$ and $\overline{r_j}$ respectively represent the mean ratings of user $c_a$ and user $c_b$, respectively. $r_{a,k}$ and $r_{b,k}$ are the ratings of user $c_a$ and $c_b$ for good $I_k$. $\varphi_a$ and $\varphi_b$ respectively represent the standard deviations for user $c_a$ and $c_b$, and the calculation method is shown by Equation (7) below:

$$\varphi_a = \sqrt{\sum_{k \in S_{a,b}} \left(r_{j,a} - \overline{\overline{r_i}}\right)^2}$$
$$\varphi_b = \sqrt{\sum_{k \in S_{a,b}} \left(r_{j,b} - \overline{\overline{r_j}}\right)^2} \tag{7}$$

$$Sim_{a,b} = \delta_1 Sim1_{a,b} + \delta_2 Sim2_{a,b} + \delta_3 Sim3_{a,b}, \tag{8}$$

where, $\delta_i$ represents the weight index of different similarity, respectively. We can get the similarity degree between users $c_a$ and $c_b$ according to Equation (8) (the higher, the better). The set of similar users for the target user $c_a$ is defined as $Sim_{c[a]}$ and the threshold number of similar users is set as q in the experiment. Thus, we defined the dataset of similar users for the target user $c_a$ as $Sim_{c[a]} = (c_1, c_2, \ldots, c_n)$.

### 3.2. Establishment of the Recommendation Model

### 3.2.1. The Recommendation Probability for Each Good According to the Historical Purchase Records

To recommend goods to users more effectively, we needed to calculate the recommendation probability of each merchandise on the basis of the purchase records of users. The main calculation methods are illustrated below:

Suppose that the past states are $V_0 = x_0, V_1 = x_1, \ldots, V_{t-1} = x_{t-1}$ and that the present state is $V_t = x_t$, where $V_t = x_t$ represents the state being $x_t$ at time t; the value of $x_t$ is 0 or 1. In this case, the state probability at the next time step $x_{t+1}$ is represented by Equation (9):

$$p(V_{t+1} = x_{t+1} | V_t = x, V_t = x_{t-1}, \ldots, V_{t-m+1} = x_{t-m+1}), \tag{9}$$

where $p$ represents the probability of the state at the next time. Therefore, we can obtain the probability recommendation matrix of both the existing store for reference and the new store for the target user. For example, in order to get the recommendation probability matrix of the reference existing store for target user $c_a$, the transfer matrix can be illustrated as Equation (10) below:

$$H_{u_a}^e = \begin{bmatrix} g_{1,1}^e & g_{1,2}^e & \cdots & g_{1,n}^e \\ g_{2,1}^e & g_{2,2}^e & \cdots & g_{2,n}^e \\ \vdots & \vdots & \ddots & \vdots \\ g_{n,1}^e & g_{n,2}^e & \cdots & g_{n,n}^e \end{bmatrix}, \tag{10}$$

where $H_{c_a}^e$ represents the recommendation probability matrix of each piece of merchandise for the target user, and $e$ represents the purchase records in the typical store for similar users and the target user. $g_{a,b}^e$ represents the probability that the target user will purchase good $I_j$ at the next time instant $t$ + 1 in the condition that the historical purchase records are $e$ and the target user has purchased good $I_i$ at the current time $t$. The calculation method of $g_{a,b}^e$ is shown as Equation (11) below:

$$g_{a,b}^e = p\left(i_{t+1} \in B_{t+1}^{c_b} | a \in B_t^{c_b}\right) = \frac{p\left(a_{t+1} \in B_{t+1}^{c_b} \wedge a \in B_t^{c_b}\right)}{p\left(a \in B_t^{c_a}\right)} = \frac{num(c(t \to t+1))}{num(c(t))}, \tag{11}$$

where $B_{t+1}^{c_a}$ represents the set of goods that the user $c_a$ will purchase at time $t + 1$ and $B_t^{c_a}$ indicates the set of goods that user $c_a$ has purchased at time $t$. $num(c(t))$ represents the number of users that have purchased good $I_i$ at time $t$, and $num(c(t \rightarrow t + 1))$ represents the number of users that have purchased good $I_i$ at time $t$ and purchased product $I_j$ at time $t + 1$. At the beginning of the experiment, the newly opened store has almost no historical purchase records; therefore, it is difficult to find similar users for the target user $c_a$. Along with the experimental training, the purchase records of the new store are gradually increasing, and we can therefore search for similar users to the target user, and the calculation method is the same as above. For instance, assume that there is a target user $c_a$ and that similar users can first be obtained based on the purchase records in the reference existing store; after that, we can get the recommendation probability matrix of the target user $c_a$. Suppose the threshold $n = 4$, the time window $w = 3$ and the number of good 5. The historical purchase records of similar users in the reference existing store when the time window $m \leq 3$ are shown below:

$$
k_1 : \begin{bmatrix} 0 & 1 & 0 \\ 0 & 0 & 0 \\ 1 & 0 & 1 \\ 0 & 1 & 0 \\ 0 & 0 & 1 \end{bmatrix}
k_2 : \begin{bmatrix} 0 & 0 & 1 \\ 1 & 0 & 0 \\ 0 & 1 & 0 \\ 1 & 0 & 1 \\ 0 & 1 & 0 \end{bmatrix}
k_3 : \begin{bmatrix} 0 & 0 & 0 \\ 1 & 0 & 1 \\ 0 & 1 & 0 \\ 1 & 0 & 0 \\ 1 & 0 & 0 \end{bmatrix}
k_4 : \begin{bmatrix} 0 & 0 & 1 \\ 1 & 0 & 0 \\ 0 & 0 & 0 \\ 0 & 1 & 0 \\ 0 & 1 & 0 \end{bmatrix},
$$

where the row of matrix $k_i$ represents the number of goods and the column of matrix $k_i$ represents the case of historical purchased $X_j$. If a user has purchased good $I_3$ at time 2, the result of row 3 and column 2 is 1. Otherwise, the result of row 3 and column 2 is 0.

The historical purchase records of the target user are shown below:

$$
k : \begin{bmatrix} 0 & 1 & 0 \\ 1 & 0 & 0 \\ 1 & 0 & 0 \\ 0 & 0 & 1 \\ 0 & 1 & 0 \end{bmatrix}.
$$

According to Equation (10), the result of recommendation probability matrix of $c_a$ can be computed below:

$$
H_{u_a}^e = \begin{bmatrix} 0 & 0 & 0.25 & 0.25 & 0.25 \\ 0.25 & 0 & 0.5 & 0.25 & 0.75 \\ 0.75 & 0.25 & 0 & 0.5 & 0.25 \\ 0.2 & 0 & 0.6 & 0 & 0.4 \\ 0.4 & 0 & 0.2 & 0.4 & 0 \end{bmatrix}.
$$

In order to explain the results above, we computed the result of g3,1 as an illustration. It is clear that the number of users between both the similar users and the target user that have ever purchased $I_3$ it is 5 at time $t$. Thus, the denominator of result is 5. In the case that users have purchased good $I_3$, the number of users that purchase good $I_1$ is 1 at time $t + 1$. Thus, the numerator of result is 1. Therefore, we can get the result for g3,1 is 0.2.

### 3.2.2. The Calculation Method for the Correlation Relationships between Goods

However, it is necessary to consider the correlation relationships among goods in recommendation systems. According to the characteristics of goods and the categories they belong to, the relationships among goods are taken into consideration on the basis of the information flow on the Internet, which means that if the flow of information is larger, the correlation relationship among the items is closer.

In our paper, $S = (s_{i,j})$ is defined as the correlation relationship between goods in the scheme, where $s_{i,j}$ represents the probability of the correlation relationship of good $I_i$ and good $I_j$. According to the definition of $s_{i,j}$, it can be seen that the value of $s_{i,j}$ is in the interval (0, 1).

The matrix of the correlation relationship of goods is illustrated as follows:

$$S = \begin{bmatrix} s_{1,1} & s_{1,2} & \cdots & s_{1,n} \\ s_{2,1} & s_{2,2} & \cdots & s_{2,n} \\ \vdots & \vdots & \ddots & \vdots \\ s_{n,1} & s_{n,2} & \cdots & s_{n,n} \end{bmatrix}. \tag{12}$$

After that, we calculated the result of each $S_{i,j}$. $B_{i,j}$ is defined as the number of users that have bought both good $I_i$ and good $I_j$. The computing method of $S_{i,j}$ is illustrated as follows:

$$S_{i,j} = \lambda_{i,j}^{S} \times h(B_{i,j}) = \lambda_{i,j}^{S} \times \frac{1}{1 + e^{-B_{i,j}}}, \tag{13}$$

where $\lambda_{i,j}^{S}$ represents whether there is a relationship among good $I_i$ and good $I_j$. If there is a relationship among $I_i$ and $I_j$, the result of $\lambda_{i,j}^{S}$ is 1. Otherwise, the result of $\lambda_{i,j}^{S}$ is 0. $h(B_{i,j})$ is a logical function that can qualify the result of $S_{i,j}$ during interval [0, 1]. It can be seen that the result of $S_{i,j}$ is symmetric, which means that the value of $S_{i,j}$ is equal to that of $S_{j,i}$.

We suppose that there is a correlation relationship between $I_1$ and $I_2$ and between $I_3$ and $I_5$. If $B_{1,2} = 4$ and $B_{3,5} = 2$, then we can obtain the results that $s_{1,2} = s_{2,1} = 0.892$, $s_{3,5} = s_{5,3} = 0.889$, and others in the recommendation probability matrix $S$ are 0. The results for the recommendation matrix of the relationship correlations between goods are shown below:

$$S = \begin{bmatrix} 0 & 0.892 & 0 & 0 & 0 \\ 0.892 & 0 & 0 & 0 & 0 \\ 0 & 0 & 0 & 0 & 0.889 \\ 0 & 0 & 0 & 0 & 0 \\ 0 & 0 & 0.889 & 0 & 0 \end{bmatrix}.$$

### 3.2.3. The Mean Recommendation Probability Matrix of Goods

Then, the combination recommendation probability matrix can be obtained in the reference existing store, and the calculation method is as follows:

$$A_{c_i}^{e} = h * H_{c_i}^{e} + (1 - h) * S, \tag{14}$$

where $A_{c_i}^{e}$ represents the final recommendation probability matrix and $h$ represents the influence factor. The recommendation probability matrix is as shown below:

$$A_{c_i}^{e} = \begin{bmatrix} b_{1,1}^{e} & b_{1,2}^{e} & b_{1,3}^{e} & b_{1,4}^{e} & b_{1,5}^{e} \\ b_{2,1}^{e} & b_{2,1}^{e} & b_{2,3}^{e} & b_{2,4}^{e} & b_{5,1}^{e} \\ b_{3,1}^{e} & b_{3,2}^{e} & b_{3,3}^{e} & b_{3,4}^{e} & b_{3,5}^{e} \\ b_{4,1}^{e} & b_{4,2}^{e} & b_{4,3}^{e} & b_{4,4}^{e} & b_{4,5}^{e} \\ b_{5,1}^{e} & b_{5,2}^{e} & b_{5,3}^{e} & b_{5,4}^{e} & b_{5,5}^{e} \end{bmatrix}, \tag{15}$$

where $b_{i,j}^{e}$ represents the recommendation probability of each good after adding the factor of the correlation relationship between goods into $H_{c_i}^{e}$.

According to Equation (14), if $h = 0.6$, according to the results of $H_{u_i}^{e}$ and $S$, we can get the results of the final probability recommendation matrix $A_{c_i}^{e}$ with the data of similar users and the correlation

relationships among goods in the experiment. The probability recommendation matrix $A^e_{c_i}$ is shown as follows:

$$A^e_{u_i} = \begin{bmatrix} 0 & 0.36 & 0.15 & 0.15 & 0.15 \\ 0.51 & 0 & 0.3 & 0.15 & 0.45 \\ 0.45 & 0.15 & 0 & 0.3 & 0.51 \\ 0.12 & 0 & 0.36 & 0 & 0.24 \\ 0.24 & 0 & 0.48 & 0.24 & 0 \end{bmatrix}.$$

Therefore, on the basis of the matrix of analysis above, we can obtain the mean transition probability of each good in the reference existing store. The calculated method is denoted as Equation (16):

$$p\left(i_{t+1} \in B^{ca}_{t+1}\right) = \frac{1}{\left|B^{ca}_t\right|} \cdot \sum_{a \in B^{ua}_t} p\left(i_{t+1} \in B^{ca}_{t+1} \middle| B^{ca}_t\right), \tag{16}$$

where $\left|B^{ca}_t\right|$ represents the number of goods that the target user has bought at time $t$. Based on the above example and Equation (16), the final recommendation probability of each good in the existing store is:

$$p\left(I_1 \in B^{ca}_{t+1} \middle| I_4\right) = \frac{1}{1} \cdot 0.24 = 0.12$$

$$p\left(I_2 \in B^{ca}_{t+1} \middle| I_4\right) = \frac{1}{1} \cdot 0 = 0$$

$$p\left(I_3 \in B^{ca}_{t+1} \middle| I_4\right) = \frac{1}{1} \cdot 0.24 = 0.36$$

$$p\left(I_4 \in B^{ca}_{t+1} \middle| I_4\right) = \frac{1}{1} \cdot 0 = 0$$

$$p\left(I_5 \in B^{ca}_{t+1} \middle| I_4\right) = \frac{1}{1} \cdot 0 = 0.24.$$

When a new store opens, although it does not have purchase records, the relationships between goods can be determined by referring to those of the existing store for reference.

### 3.2.4. The Trust Factor of Goods in the RM-MES Scheme

In traditional recommendation schemes, there exists a dependency among users in social networks. If two users have a similar performance, the trust level is obviously high. Therefore, in this paper, the trust factor is added into the RM-MES scheme to improve the accuracy of recommendation results. In the RM-MES scheme, the trust factor of a good is divided into the reputation, sales rank, and frequency. The calculated method of trust is denoted as follows:

$$trust_i = \tau \cdot rep_i + \theta \cdot \frac{1}{e^{rank_i}} + (1 - \tau - \theta) \cdot \frac{fre_i}{Fre}, \tag{17}$$

where $trust_i$ represents the trust degree of good $i$, $rep_i$ represents the reputation of good $I$, and $fre_i$ represents the purchased frequency of good $i$. $Fre$ is a constant in the experiment, which makes sure that the value of $fre_i/Fre$ is in the interval [0, 1]. $\tau$ and $\theta$ respectively represent the scale factor of reputation and the influence factor of sales rank for good $I_i$. $rank_i$ is the sales rank of good $I_i$.

Because the historical purchase records are rich in the existing store for reference, the value of $rep_i$, $rank_i$, and $fre_i$ for good $i$ is certain. With the operation of the RM-MES scheme, the reputation, sales rank, and purchase frequency in the recently opened store change at different time cycles.

### 3.2.5. The Latent Factors of Users in the RM-MES Scheme

In the existing store, it is easy to determine the target user's transition matrix of probability by the histories of browsing and trust factor of goods if the target user is not new. However, there are few histories of browsing and trust degrees of goods for a recently opened store, which is called a

"cold start". In our paper, we defined $L_a = (age, gender, location, browse)$ as the attribute set of latent factors. If the target user is new, and there is therefore no historical purchase record, it is not easy to recommend accurate goods for the user. However, the set of latent similar users can be adopted to compute the latent goods that the target user may like. The set of similar users for the target user $c_a$ is defined as $Sim(Latent(c_a), Latent(c_a))$, which can be computed by the four factors shown in the Equation (18).

$$Sim(Latent(c_a), Latent(c_b))$$
$$= \delta_1 \cdot Sim(Age(c_a), Age(c_b)) + \delta_2 \cdot Sim(Gender(c_a), Gender(c_b)) \qquad , \qquad (18)$$
$$+ \delta_3 \cdot Sim(Location(c_a), Location(c_b)) + \delta_4 \cdot Sim(Browse(c_a), Browse(c_b))$$

where $\delta$ represents the influence factor of attribute similarity, $\delta_1 + \delta_2 + \delta_3 + \delta_4 = 1$, $Sim(Age(c_a), Age(c_b))$ represents the latent similarity relationship of age between $c_a$ and $c_b$, $Sim(Gender(c), Gender(c_b))$ represents the latent similarity of gender between $c_a$ and $c_b$, $Sim(Location(c_a), Location(c_b))$ represents the latent similarity of location between $c_a$ and $c_b$, and $Sim(Browse(c_a), Browse(c_b))$ represents the latent similarity of browsing between $c_a$ and $c_b$.

### 3.2.6. The Establishment of Combination Calculation

Based on the methods shown above, the RM-MES scheme combines the mean recommendation probability matrix of goods for target users, trust degree of selected goods, and latent factor of target users, to establish the computation method illustrated below:

$$R_{c_a}^{I_j} = x \cdot p\left(i_{t+1} \in B_{t+1}^{c_i}\right) + y \cdot trust_i + z \cdot latent_{c_a}^{I_j}, \ x + y + z = 1, \qquad (19)$$

where $R_{c_a}^{I_j}$ represents the recommendation probability in the existing selected store to recommend good $I_j$ to target user $c_a$. $x$ and $y$ represent, respectively, the weight of the mean probability matrix of recommended goods for user $c_a$ and the trust degree for good $I_j$, and $z$ is the weight when the target user $c_a$ is new. If the target user selected is new, then the historical purchase records are empty, and therefore $x + y = 1$, $z = 0$.

In the RM-MES scheme, we calculated $R_{c_a}^{I_j}$ in both the existing store and the new store and then combine them together to recommend appropriate goods for the target users. In the new shop, the recommendation probabilities of goods are defined as $R_{1c_a}^{I_j}$. The calculation method of $R_{1c_a}^{I_j}$ is the same as that in $R_{c_a}^{I_j}$. The calculation method for the combination of $R_{1c_a}^{I_j}$ and $R_{c_a}^{I_j}$ can be obtained by Equation (20) as follows:

$$R_{fc_a}^{I_j} = \mu \cdot R_{1c_a}^{I_j} + (1 - \mu) \cdot R_{c_a}^{I_j}, \qquad (20)$$

where $R_{fc_a}^{I_j}$ represents the recommendation probability of providing good $I_j$ to user $c_a$, and $\mu$ is the influence factor of historical purchase records in the recently opened store. The historical purchase records of a recently opened store are sparse, and therefore the result of $R_{1c_a}^{I_j}$ for the recently opened store is almost 0. Thus, at the beginning of our experiment, the value of the influence factor $\mu$ was zero; with the running of the recently opened store, the historical purchase records in the recently opened store will grow larger, and the value of $\mu$ will increase. We can calculate the value of the influence factor $\mu$ by Equation (21) as follows:

$$\mu = \frac{\sum_{i=1}^{n} fre_i}{Fre}, \qquad (21)$$

where $\sum_{i=1}^{n} fre_i$ represents the sum of the purchase frequencies of goods $I_1$ to $I_n$ in a time period and $Fre$ is a constant number, which was defined above. With the operation of the recently opened store, the historical purchase record will increase; therefore, the influence factor $\mu$ will become larger. When $\sum_{i=1}^{n} fre_i$ reaches the threshold total purchase number $n$, the store can recommend goods to users on the basis of its own historical purchase records.

The pseudo-code of the RM-MES algorithm is shown in Algorithm 1.

---

**Algorithm 1.** The main RM-MES Algorithm

---

Input: $S, I, I_i, n, m, d, q, a[i][j][k], a_1[i][j][k], fre[i], fre_1[i], x, y, z, \mu, \omega_i, \delta_i$

Output: $R_{fc_a}^{I_j}$

1: for each $s_i \in S$

2:　$R[i] = \frac{num(I) \cap num(I_i)}{num(I) \cup num(I_i)}, ref = \max(R[i],\ ref)$

3: end for

4: for each $c_i \in C$

5:　for each $c_{ij} \in c_i$

6:　　$Score = \sum_j^m (-1)^t * k * word(d)$

7:　end for

8:　$Rep_i = \quad \omega_1 Score(quality) + \omega_2 Score(price) + \omega_3 Score(appearance) + $
　　　　　　$\omega_4 Score(configuration) + \omega_5 Score(service) + \omega_6 Score(delivery)$

9:　Calculate $Sim_{a,i}$ according to Equations (4)–(8);

10: end for

11: Reverse order by $Sim_{a,i}$ and obtain $Sim_{c[a]} = (c_1, c_2, \ldots, c_q)$;

12: for each $c_i \in Sim_{c[a]}$

13:　Calculate the transfer matrix $H_{c_a}^e$ according to Equations (10) and (11);

14:　Calculate the transfer matrix $S$ based on the relationship:

15:　$S_{a,i} = \lambda_{a,i}^S \times g(B_{a,i}) = \lambda_{a,i}^S \times \frac{1}{1 + e^{-B_{a,i}}}$

16: end for

17: Calculate the final transfer matrix $A_{c_a}^e$ based on $H_{c_a}^e$ and $S$:

18: $A_{c_a}^e = h * M_{c_a}^e + (1 - h) * S$

19: Calculate the recommendation probability at the next time instance based on the historical purchase records a[i][j][k] of users:

20: $p\left(i_{t+1} \in B_{t+1}^{c_a}\right) = \frac{1}{|B_t^{c_a}|} \cdot \sum_{i \in B_t^{c_a}} p\left(i_{t+1} \in B_{t+1}^{c_a} \middle| B_t^{c_a}\right)$

21: for each $r_i \in ref$

22:　Calculate the trust degree of each good:

23:　$trust_i = \tau \cdot rep_i + \theta \cdot \frac{1}{e^{rank_i}} + (1 - \tau - \theta) \cdot \frac{fre_i}{Fre}$

24:　Calculate the latent factor if the target user is new;

25:　Comprehensively compute the probability:
　　$R_{c_a}^{I_j} = x \cdot p\left(i_{t+1} \in B_{t+1}^{c_a}\right) + y \cdot trust_i + z \cdot latent_{c_a}^{I_j}$

26:　Combine the recommendation probabilities of the reference shop and new shop:

27:　$R_{fc_a}^{I_j} = \mu \cdot R_{1c_a}^{I_j} + (1 - \mu) \cdot R_{c_a}^{I_j}$

28: end for

29: Return $R_{fc_a}^{I_j}$

---

## 4. Experimental Evaluations and Results

### 4.1. Experimental Settings

In order to evaluate the effectiveness and performance of the RM-MES scheme, the purchasing network metadata of Amazon products (http://jmcauley.ucsd.edu/data/amazon/) and user review information were used in our experiments [11,15]. First, to evaluate the effectiveness and performance of the RM-MES scheme, we closely compared the RM-MES scheme with the classic trust-based scheme under a certain influence factor in different time periods. Then, we verified the influence factor of the influence factor x in the RM-MES scheme. In addition, we compared the trust degree of each selected good under different time periods. Finally, we compared and analyzed various detailed results during the experiment.

The dataset in the experiment was obtained by enquiring into the dataset on the Amazon website. We chose the metadata and reviews of the health and personal care category, which contains approximately 263,032 different goods. For each user, the following information could be obtained:

ID of the product bought, the review ID, comment on the product, and review time. For each piece of merchandise, the following information could be obtained: ID, sales rank, categories, description, and list of similar goods. From the health and personal care catalogue, in our experiment, we chose several kinds of goods with higher purchase ranking in the experiments. In our experiments, we used 3/4 of the selected purchase records as the training set and the rest as the test set.

Then, to evaluate the effectiveness and performance of the RM-MES scheme, we compared results of precision, recall, and $F_{1\text{-measure}}$. The three indexes above are three standard measurements for measuring the effectiveness of a recommendation scheme (the higher, the better). The recall can be obtained by the following calculation method (Equation (22)):

$$precision = \frac{1}{H} \cdot \sum_{a=1}^{H} \frac{N_a}{List_a}, \tag{22}$$

where $H$ represents the total number of both the target user and similar users, $N_a$ indicates the number of goods that user $c_a$ purchased in the recommendation list, and $List_a$ indicates the number of goods in the recommendation list.

The recall in the RM-MES scheme can be computed by Equation (23) as follows:

$$recall = \frac{1}{H} \cdot \sum_{a=1}^{H} \frac{N_a}{B_a}, \tag{23}$$

where $B_a$ represents the number of goods that user $c_a$ likes on the basis of the comments given by user $c_a$ and recall indicates the number of goods that target user $c_a$ likes in the list of recommendation to the total number of goods that user $c_a$ likes. The bigger the value of recall, the better.

Because $F_{1\text{-measure}}$ is calculated as a combination of these two indicators, $F_{1\text{-measure}}$ can comprehensively verify the effectiveness of the RM-MES scheme. The calculation method of $F_{1\text{-measure}}$ can be obtained by Equation (24) as follows:

$$F_{1-measure} = \frac{2 \cdot recall \cdot precision}{precision + recall}. \tag{24}$$

If the $F_{1\text{-measure}}$ of the recommendation scheme is higher, the performance of the recommendation scheme is better.

*4.2. Experimental Results*

In this section, the performance of the RM-MES scheme is compared with that of the trust-based scheme proposed in Reference [1]. The trust-based scheme is to recommend appropriate goods to users based on the trust factor of goods. We chose the health and personal care shop as the reference of the recently opened shop.

The users selected in the experiments were not new, so the latent factor of users was not considered in our experiment. In other words, $x + y = 1$ and $z = 0$. As shown in Figure 1a, the precision of the trust-based scheme was lower than that of the RM-MES scheme on average. When a new store opens, historical purchase records are more likely to be sparse, and therefore the precision is low (cold start). The RM-MES scheme had a better performance than the trust-based model on average. When time is 8, the recommendation results of precision of the newly opened store are higher than those of the existing store, which means the store can recommend goods to users with its own historical purchase records.

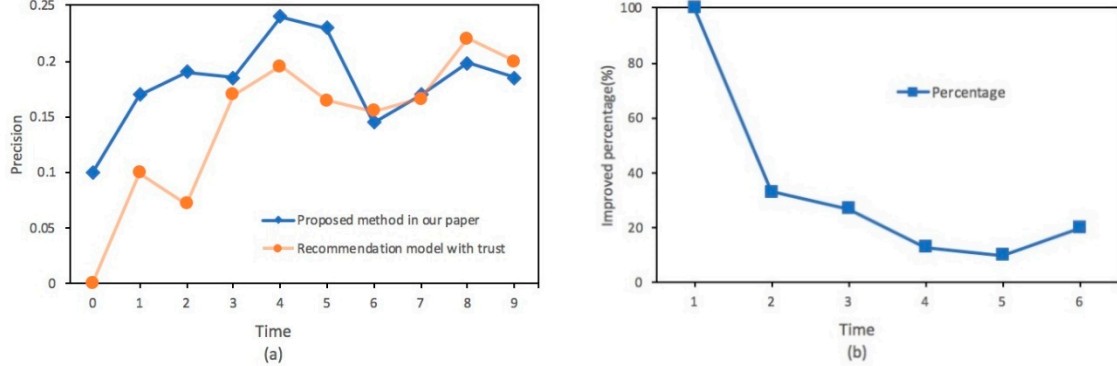

**Figure 1.** The result of the experiment. (**a**) The results of precision in a health and personal care shop; (**b**) the percentage of improvement of precision in the health and personal care shop.

From the experimental results of precision, it is clear that the precision of the trust-based scheme is lower than that of the RM-MES scheme when a store is newly opened. This is because the RM-MES scheme first refers to the correlation of goods among the existing store and the new store and then guides the new store to recommend goods to users. Thus, though the newly opened store has few purchase records, the RM-MES scheme still has a better performance than other methods. Thus, it can be seen that the RM-MES scheme can solve the "cold start" problem.

With the running of the newly opened store, there are more and more purchase records in the new store, so it is more effective for it to adopt its own purchase records. Thus, after a period of time, the precision of the recommendation model will be maintained at a constant level. It is similar to other schemes that only adopt their purchase records to recommend goods. Thus, the precision ratio of the RM-MES scheme is similar to that of other schemes.

The comparison of percentages of improvement of precision when time <6 is shown in Figure 1b. It can be seen that the percentage of improvements of precision is very high at the beginning. This is because when a new shop opens, it has few historical purchase records, so it is difficult to recommend goods for target users appropriately. However, the proposed method in our paper can combine both the RM-MES scheme and the trust-based recommendation model to recommend goods to target users. Thus, the percent improvements of precision are very high.

Then, we closely compared the result of recall for the two schemes with time passing, and the results are shown in Figure 2a. Figure 2b shows the comparison of percent improvements of recall in the RM-MES scheme. From the results of recall, it is clear that the RM-MES scheme has better effectiveness and performance than the trust-based scheme at the beginning. When a new store opens, the historical purchase records is more likely to be 0%; thus, it is difficult to recommend goods for target users appropriately (cold start). However, the existing store for reference has enough purchase records to recommend goods to target users. Therefore, the results of the recall that is the combination of both the existing store and the new store are definitely higher than those of the results of recall that only adopt their own purchase records of a new store. However, there were fluctuations during the experiment, as shown in the Figure 2a. This is because there are uncertainties in online social networks, which may cause the recall of recommendation to have fluctuations.

Then, the results of $F_{1\text{-measure}}$ of these two schemes are compared to comprehensively evaluate the performance of the proposed recommendation scheme. The results of $F_{1\text{-measure}}$ are shown in Figure 3a. Figure 3b shows the comparison of the percentage of improvements of $F_{1\text{-measure}}$ in the RM-MES scheme. From the results of $F_{1\text{-measure}}$, it is clear that the proposed method in our paper has a better performance than the trust-based scheme at the beginning. This is because the precisions and recalls of the proposed method in our paper are higher than those of other methods because the purchase records are more likely to be sparse when a new store opens. Therefore, the $F_{1\text{-measure}}$ of the proposed method is higher than that of the other methods. There is an increasing number of purchase

records in the newly opened store as time passes. The results for the $F_{1\text{-measure}}$ of the recommendation model will be maintained at a constant level.

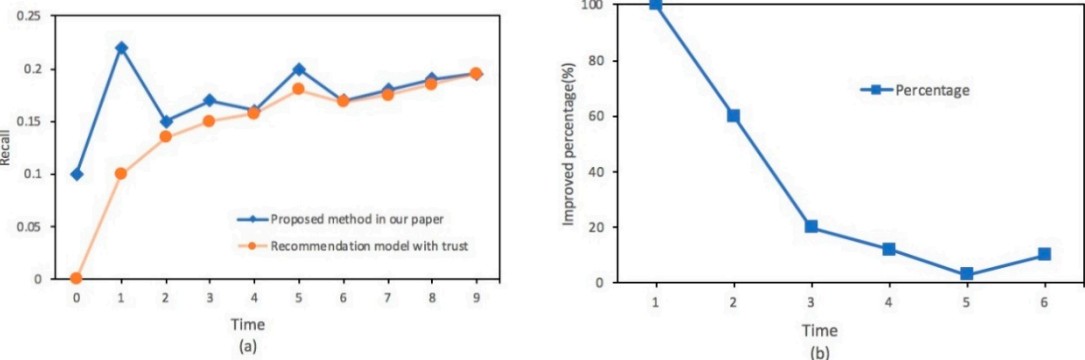

**Figure 2.** The result of the experiment. (**a**) The results of recall in a health and personal care shop; (**b**) the percentage of improvement of recall in the health and personal care shop.

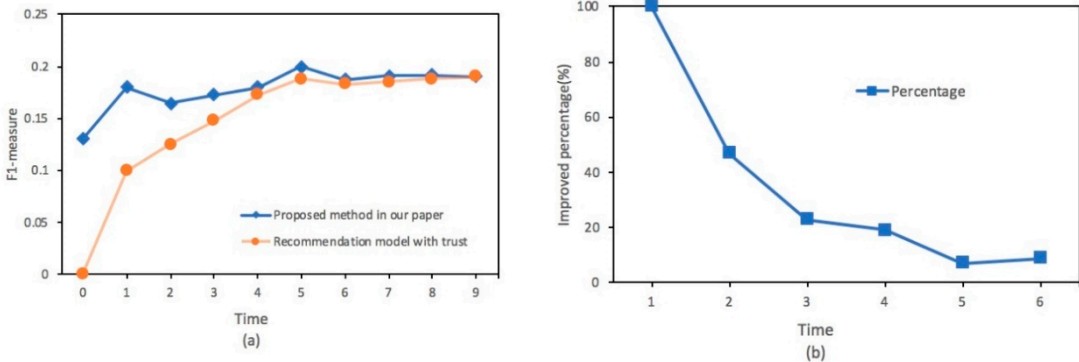

**Figure 3.** The result of the experiment. (**a**) The results of $F_{1\text{-measure}}$ in a health and personal care shop; (**b**) the percentage of improvement of $F_{1\text{-measure}}$ in the health and personal care shop.

The precisions of the two schemes are closely compared under different recommendation thresholds, as shown in Figure 4a. From the picture above, we can reach the conclusion that the effectiveness of the proposed method in our paper is better than that of the trust-based model under different recommendation thresholds $\alpha$. This is because the trust-based model selects their own purchase records to recommend. However, the purchase matrixes are more likely to be empty at the beginning. Thus, it is difficult to recommend goods for target users appropriately. When the recommendation threshold is 0.5, the results of precision in our proposed method are the highest and decrease over time, and they remain at zero when the threshold is 0.9.

A comparison of percentages of improvements of precision under different recommendation thresholds is shown in Figure 4b. As illustrated above, it can be seen that the percentage of precision is further improved in the RM-MES scheme proposed in this paper.

The recalls of the RM-MES scheme proposed in this paper and the trust-based scheme under different recommendation thresholds are shown in Figure 5a. From Figure 5a, it can be seen that the recall for the RM-MES scheme proposed in this paper is greater than that of the trust-based model under different recommendation thresholds. This is because the datasets of the RM-MES scheme are a combination of the historical purchase records of the existing store and the historical purchase records of the target store. When the recommendation threshold $\alpha$ is higher than 0.4, the results of recall in the RM-MES scheme gradually become smaller. The recalls of these two recommendation schemes remain at zero when the recommendation threshold is 0.9.

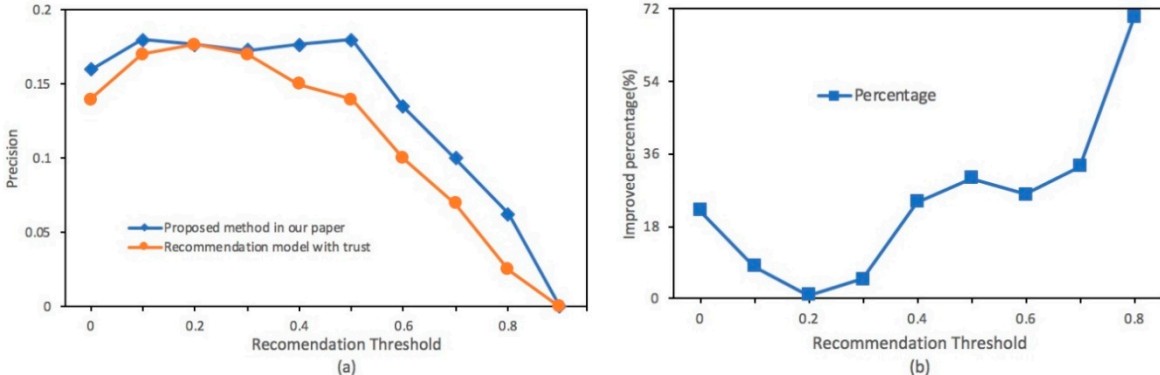

**Figure 4.** The result of the experiment in the health and personal care shop. (**a**) The results of precision under different recommendation thresholds; (**b**) the percentage of improvement of precision under different recommendation thresholds.

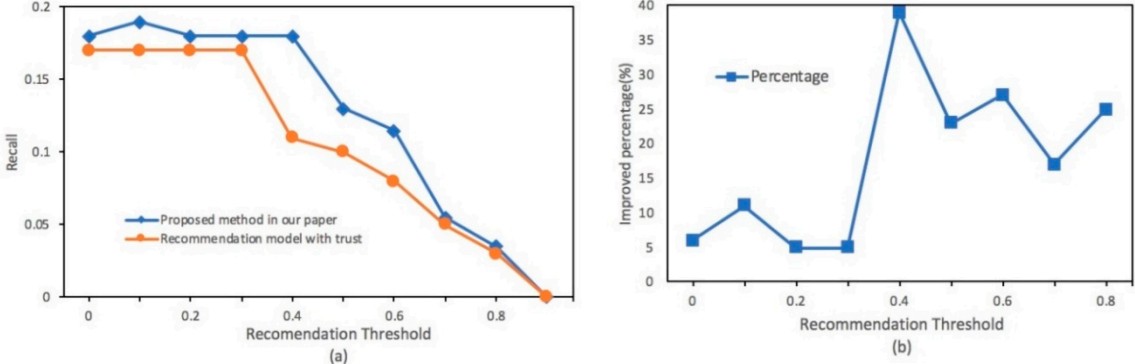

**Figure 5.** The result of the experiment in the health and personal care shop. (**a**) The results of recall under different recommendation thresholds; (**b**) the percentage of improvement of recall under different recommendation thresholds.

A comparison of percent improvements of recall under different recommendation thresholds is illustrated in Figure 5b.

To comprehensively evaluate the performance of the proposed scheme, the $F_{1\text{-measure}}$ results of these two recommendation schemes were compared under different recommendation thresholds. The $F_{1\text{-measure}}$ of the RM-MES scheme proposed in this paper and the trust-based scheme under different recommendation thresholds is illustrated in Figure 6a. It is clear that the effectiveness of the $F_{1\text{-measure}}$ in the RM-MES scheme is greater than that of the trust-based scheme. However, the RM-MES scheme combines both the historical purchase records of reference existing stores and newly opened stores together. Thus, at the beginning, it can recommend goods to users more accurately than the trust-based scheme. When the recommendation threshold is 0.9, the recall and precision in both a trust-based scheme and RM-MES scheme are zero; therefore, the $F_{1\text{-measure}}$ of these two recommendation schemes are both 0. In addition, it can be seen from the results below that the RM-MES scheme is more stable than the trust-based schemes. A comparison of percentages of improvements of $F_{1\text{-measure}}$ under different recommendation thresholds is illustrated in Figure 6b. From the results shown above, it can be seen that the percentages of the $F_{1\text{-measure}}$ are further improved in the RM-MES scheme under different thresholds. When the recommendation threshold is more than 0.8, the recall and precision of both the proposed scheme in our paper and the trust-based scheme are zero; therefore, $F_{1\text{-measure}}$ reaches zero in the experiment.

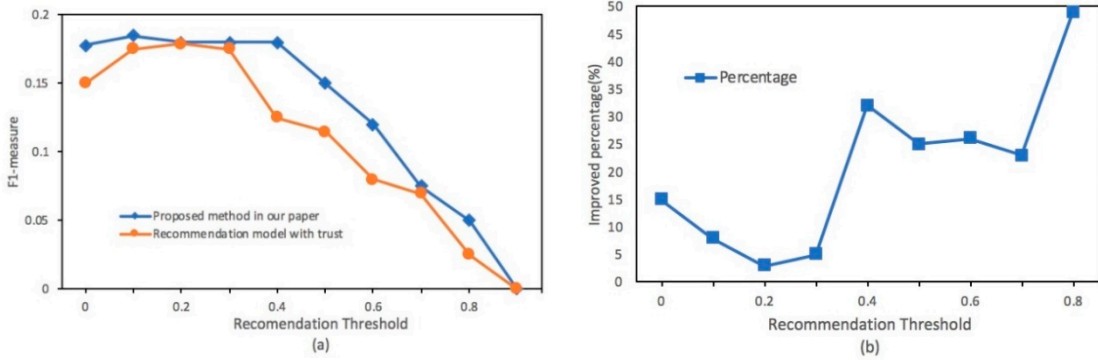

**Figure 6.** The result of the experiment in the health and personal care shop. (**a**) The results of $F_{1\text{-measure}}$ under different recommendation thresholds; (**b**) the percentage of improvement of $F_{1\text{-measure}}$ under different recommendation thresholds.

To further evaluate the performance of the RM-MES scheme, after the experiment of the health and personal care category is complete, we carried on the experiment of a newly opened baby products store. In the category of baby products, there are about 71,317 different kinds of products. In our experiment, we selected the most purchased products as the training set on the basis of the sales rank. In the experiments, we used 3/4 of the selected historical purchase records as the training set and the rest as the test set. Table 2 shows a summary of the experimental results.

**Table 2.** The experimental results of precision, recall, and $F_{1\text{-measure}}$ in the baby products shop.

|                      | TB    | $X = 0.3$ | $X = 0.4$ | $X = 0.5$ | $X = 0.6$ |
| -------------------- | ----- | --------- | --------- | --------- | --------- |
| Precision            | 0.115 | 0.140     | 0.130     | 0.131     | 0.105     |
| Recall               | 0.106 | 0,137     | 0.133     | 0.131     | 0.1       |
| $F_{1\text{-measure}}$ | 0.110 | 0.139     | 0.138     | 0.133     | 0.1       |

From Table 2, we can reach the conclusion that the effectiveness of the RM-MES scheme is greater than that of the other schemes. In addition, in the proposed scheme in our paper, we can obtain the best measurement results when the influence factor of transition probability $x$ is 0.3.

At the initiation stage, the results of precision ratio, recall ratio and $F_{1\text{-measure}}$ are improved by approximately 19.07%, 20.73%, and 21.02%, respectively, compared to the previous schemes.

## 5. Conclusions

In this paper, based on the trust-based recommendation model, we proposed a new recommendation model (RM-MES) based on multi-emotion similarity to improve the performance of the recommendation scheme and overcome the "cold start" problem. First, we divided users' behaviors into browsing goods, buying goods, and purchasing goods as well as evaluating these goods. Then, the recommendation attributes of goods were considered to obtain similarities between users and shops. Then, the most similar store was selected as the reference existing store in our experiment. Next, the recommendation probability matrix of both the existing store and the new store were calculated according to the similarity between users and target user. Finally, we adopted the Amazon product co-purchasing network metadata and commentary information to evaluate the effectiveness and performance of the RM-MES scheme through comprehensive experiments. Furthermore, we obtained the best measurement results when the influence factor of transition probability $x$ was 0.3 in our experiment. Therefore, we compared detailed information in the RM-MES scheme with that in the trust-based scheme through experiments when the influence factor of transition probability $x$ is 0.3 and analyzed the impact of the transition probability influence factors in the RM-MES scheme through experiments. Therefore, we can draw the conclusion that the RM-MES scheme has a better performance than other recommendation schemes.

For high probability goods in the RM-MES scheme, the RM-MES scheme will further enhance the recommended probability of goods that have been recommended and non-recommended goods will suffer further reductions in their recommendation probabilities. Therefore, this tendency leads to the phenomenon that the recommendation system loses the opportunity to recommend more optimized goods. In future studies, we will further research how to recommend other goods with small probabilities to users to bring higher profit to the system.

**Author Contributions:** Conceptualization, J.L. and Y.W.; formal analysis, X.Y. and Y.W.; funding acquisition, J.L.; investigation, Y.W.; methodology, J.L. and Y.W.; project administration, T.L. and Y.W..; software, T.L. and Y.W.; supervision, X.Y.; validation, Q.L.; visualization, X.Y.; writing (original draft), Q.L. and Y.W.; writing (review and editing), Y.W.

**Funding:** This research was funded by the National Natural Science Foundation of China (61272150, 61379110), and the Fundamental Research Funds for the Central Universities of Central South University (2018zzts596).

**Acknowledgments:** This work was supported in part by the National Natural Science Foundation of China (61472450, 61402165, 61702560, S1651002, M1450004), the Key Research Program of Hunan Province (2016JC2018), and the Natural Science Foundation of Hunan Province (2018JJ2099). The authors would like to thank the reviewers for their valuable suggestions and comments.

**Conflicts of Interest:** The authors declare no conflict of interest.

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
