# Peer review of "A Recommendation Model Based on Multi-Emotion Similarity in the Social Networks"

_information, doi:10.3390/info10010018_

Round 1
Reviewer 1 Report
Minor issues:
How the "RM-MEDS" stands for "recommendation model based on multi-emotion similarity for social networks"? Where does the "D" come from?
The valid way to enter a reference number is right after the authors'/algorithms' reference in the text and not on the end of the sentence. e.g. "Y. Wang et al designed a combination model composed of the recommender and the similarity measure [6]." -> "Y. Wang et al [6] designed a combination model composed of the recommender and the similarity measure."
Before listing items, you have to use a ":" (related work, line 4).
In algorithm 1 (page 13) it is extremely hard to follow the idea. Please use tabs in order to show which commands are in an if clause and/or for loop etc.
"The purpose of this paper" is usually written in abstract, introduction and conclusion and not in the algorithm's proposal/presentation.
Major issues:
I see a detailed papers' reference in introduction and a poor related work section.The pattern "X et al. ..." is usually a related work's pattern and not an introduction's pattern. Hence, please try to follow the concept that the introduction "introduces" the paper's idea to the reader, otherwise the reader will be lost between your work and other people's (your references) work.
Since you use the Amazon datasets, you must give a reference, otherwise copyright issues might come up (http://jmcauley.ucsd.edu/data/amazon/).
Throughout the text, the quality of the English language should be improved.
In your experimental results, you compared your work only with the work proposed in [1] (Y. Wang, G. Yin, Z. Cai, Y. Dong, and H. Dong, "A trust-based probabilistic recommendation model for social networks"). Is this 3 year old paper the only state-of-the-art paper in this area? I can find a lot of works in this area and neither reference nor comparison (at least mention why a comparison with these works is not applicable):
Choudhary N., Bharadwaj K.K. (2019) Leveraging Trust Behaviour of Users for Group Recommender Systems in Social Networks
Contratres F.G., Alves-Souza S.N., Filgueiras L.V.L., DeSouza L.S. (2018) Sentiment Analysis of Social Network Data for Cold-Start Relief in Recommender Systems
Personalized recommender system based on friendship strength in social network services, YD Seo, YG Kim, E Lee, DK Baik
Exploiting emotion on reviews for recommender systems X Meng, S Wang, H Liu, Proc. AAAI Conf 2018
Author Response
Dear Editors and Reviewers,
We do appreciate your valuable suggestions on our paper entitled “A Recommendation Model based on Multi-emotion Similarity in the Social Networks (Manuscript ID: information-407433). After receiving the suggestions, we have made great efforts to carefully revise the manuscript according to the comments and suggestions of reviewers. The responses to reviewers are as follows:
Minor issues:
Point 1: How the "RM-MEDS" stands for "recommendation model based on multi-emotion similarity for social networks"? Where does the "D" come from?
Response 1: Thank you for your advices, we have revised this mistake in the new submission. The abbreviation of "recommendation model based on multi-emotion similarity for social networks" was incorrectly stated in the original manuscript. In the revised paper, we use "RM-MES" to stand for the "recommendation model based on multi-emotion similarity for social networks".
Point 2: The valid way to enter a reference number is right after the authors'/algorithms' reference in the text and not on the end of the sentence. e.g. "Y. Wang et al designed a combination model composed of the recommender and the similarity measure [6]." -> "Y. Wang et al [6] designed a combination model composed of the recommender and the similarity measure."
Response 2: Thank you for your advices, we have carefully revised the whole paper word by word to avoid any mistakes.
Point 3: Before listing items, you have to use a ":" (related work, line 4).
Response 3: Thank you for your advices, it is our negligence when writing this paper. In the revised paper, we have added the “:” before listing items. More rigorous attitude will be taken when revising the paper.
Point 4: In algorithm 1 (page 13) it is extremely hard to follow the idea. Please use tabs in order to show which commands are in an if clause and/or for loop etc.
Response 4: Thank you for your advices. We have rewritten the algorithm 1 in the page 13 to make readers easy to follow. We do hop that the revised paper can meet the standards of the Information Journal. The revised algorithm is written in red.
Point 5: "The purpose of this paper" is usually written in abstract, introduction and conclusion and not in the algorithm's proposal/presentation.
Response 5: Thank you for your suggestions. We have moved "The purpose of this paper" to the introduction section in the revised paper. The revised portions are underlined in red.
Major issues:
Point 6: I see a detailed papers' reference in introduction and a poor related work section. The pattern "X et al. ..." is usually a related work's pattern and not an introduction's pattern. Hence, please try to follow the concept that the introduction "introduces" the paper's idea to the reader, otherwise the reader will be lost between your work and other people's (your references) work.
Response 6 Thank you for your suggestions. In the revised paper, we have carefully revised the introduction section and the related-work section, which are rewritten with the red color. We do hope that the revised paper can satisfy the strands of the Information Journal.
Point 7: Since you use the Amazon datasets, you must give a reference, otherwise copyright issues might come up (http://jmcauley.ucsd.edu/data/amazon/).
Response 7: Thank you for your advices, we provided a reference of the Amazon datasets in the revised paper.
Point 8: Throughout the text, the quality of the English language should be improved.
Response 8: Thank you for your advices. In the revised paper, we have carefully checked the words and grammars word by word to avoid any mistakes. We do hope that the revised paper can meet the standards of the journal.
Point 9: In your experimental results, you compared your work only with the work proposed in [1] (Y. Wang, G. Yin, Z. Cai, Y. Dong, and H. Dong, "A trust-based probabilistic recommendation model for social networks"). Is this 3 year old paper the only state-of-the-art paper in this area? I can find a lot of works in this area and neither reference nor comparison (at least mention why a comparison with these works is not applicable):
Choudhary N., Bharadwaj K.K. (2019) Leveraging Trust Behaviour of Users for Group Recommender Systems in Social Networks
Contratres F.G., Alves-Souza S.N., Filgueiras L.V.L., DeSouza L.S. (2018) Sentiment Analysis of Social Network Data for Cold-Start Relief in Recommender Systems
Personalized recommender system based on friendship strength in social network services, YD Seo, YG Kim, E Lee, DK Baik
Exploiting emotion on reviews for recommender systems X Meng, S Wang, H Liu, Proc. AAAI Conf 2018
Response 9: Thank you for your advices. We have carefully read the papers given by the reviewers and compared those papers with the RM-MES scheme. In addition, we have added the comparisons between those papers and the RM-MES scheme to the revised paper to show the advancements of the proposed scheme. The papers which have been added to the revised paper are shown in the following:
[1] Contratres F.G., Alves-Souza S.N., Filgueiras L.V.L., DeSouza L.S. (2018) Sentiment Analysis of Social Network Data for Cold-Start Relief in Recommender Systems
[2] Personalized recommender system based on friendship strength in social network services, YD Seo, YG Kim, E Lee, DK Baik
[3] Exploiting emotion on reviews for recommender systems X Meng, S Wang, H Liu, Proc. AAAI Conf 2018
In the revised paper, with comparisons, these study still have some problems that need to be overcome:
1. Most of the recommendation schemes only consider the “cold start” problem of new users, but do not consider the “cold start” problem for a newly opened store, which will have a great impact on the recommend quality of recommendation system.
2. Some recommendation schemes search for user preferences by extracting user Facebook and Twitter data. But it is difficult to extract the user’s personal information due to issues such as permissions and technology. Additionally, because information that include user emotions are often incomplete and fuzzy, it is not easy to directly analyze the emotions in the information from Facebook and Twitter.
3 These recommendation systems based on emotion only considers positive and negative emotions, but does not consider customers' preferences in other aspects.
4. When calculating similarities of customers’ behaviors, most recommendation schemes do not take the correlation between projects into consideration.
5. Most recommendation schemes fail to consider the trust factor of each product, which may cause the recommendation system to provide distrusted items to target users.
We hope that the revised paper can satisfy the requirements of the journal.
Sincerely yours,
Authors: Jun Long, Yulou Wang, Xinpan Yuan, Ting Li and Qunfeng Liu

Reviewer 2 Report
This paper presents a trust recommendation system based multi-emotion and deep similarity recommendation model for social networks named RM-MEDS. The approach is promising for overcoming the cold-start problem. However, the write-up of paper is problematic. The paper needs a thorough and substantive editing of English (that significantly reduces word-count of paper, maybe 10-20% reduction) from a native English speaking editor who is an expert in the field of recommendation systems before it is ready to be formally accepted. I would suggest authors a) further elaborate their 'reference existing store', b) cite at least 5 more relevant, latest, and influential papers to strengthen (by re-writing) their 'related works' section, and c) add study limitations and future directions.
Author Response
Dear Editors and Reviewers,
We do appreciate your valuable suggestions on our paper entitled “A Recommendation Model based on Multi-emotion Similarity in the Social Networks (Manuscript ID: information-407433). After receiving the suggestions, we have made great efforts to carefully revise the manuscript according to the comments and suggestions of reviewers. The responses to reviewers are as follows:
Point 1: This paper presents a trust recommendation system based multi-emotion and deep similarity recommendation model for social networks named RM-MEDS. The approach is promising for overcoming the cold-start problem. However, the write-up of paper is problematic. The paper needs a thorough and substantive editing of English (that significantly reduces word-count of paper, maybe 10-20% reduction) from a native English speaking editor who is an expert in the field of recommendation systems before it is ready to be formally accepted. I would suggest authors
Response 1: Thank you for your suggestions, we have revised the whole paper word by word and changed several expressions of sentences on this manuscript. To make a smooth reading for reader we have revised some grammatical errors and expressions in the paper. We do hope that the revised paper can meet the standards of the journal.
Point 2: a) further elaborate their 'reference existing store',
Response 2: Thank you for your suggestions, we have provided more explanations of the 'reference existing store' in the revised paper. The Revised portion are underlined in red.
Point 3: b) cite at least 5 more relevant, latest, and influential papers to strengthen (by re-writing) their 'related works' section,
Response 3: Thank you for your advices. We have carefully read the papers given by the reviewers and compared those papers with the RM-MES scheme. In addition, we have added the comparisons between those papers and the RM-MES scheme to the revised paper to show the advancements of the proposed scheme. The papers which have been added to the revised paper are shown in the following:
[1] Contratres F.G., Alves-Souza S.N., Filgueiras L.V.L., DeSouza L.S. (2018) Sentiment Analysis of Social Network Data for Cold-Start Relief in Recommender Systems
[2] Personalized recommender system based on friendship strength in social network services, YD Seo, YG Kim, E Lee, DK Baik
[3] Exploiting emotion on reviews for recommender systems X Meng, S Wang, H Liu, Proc. AAAI Conf 2018
In the revised paper, with comparisons, these study still have some problems that need to be overcome:
1. Most of the recommendation schemes only consider the “cold start” problem of new users, but do not consider the “cold start” problem for a newly opened store, which will have a great impact on the recommend quality of recommendation system.
2. Some recommendation schemes search for user preferences by extracting user Facebook and Twitter data. But it is difficult to extract the user’s personal information due to issues such as permissions and technology. Additionally, because information that include user emotions are often incomplete and fuzzy, it is not easy to directly analyze the emotions in the information from Facebook and Twitter.
3 These recommendation systems based on emotion only considers positive and negative emotions, but does not consider customers' preferences in other aspects.
4. When calculating similarities of customers’ behaviors, most recommendation schemes do not take the correlation between projects into consideration.
5. Most recommendation schemes fail to consider the trust factor of each product, which may cause the recommendation system to provide distrusted items to target users.
We hope that the revised paper can satisfy the requirements of the journal.
Point 4: c) add study limitations and future directions.
Response 4: Thank you for your advices, it is our negligence when writing this paper. In the revised paper, we have added study limitations and future directions.
Sincerely yours,
Authors: Jun Long, Yulou Wang, Xinpan Yuan, Ting Li and Qunfeng Liu

Round 2
Reviewer 1 Report
First of all, I would like to thank the authors for the major changes.
Major issues:
Please, move the paragraph with the references 6-10 to the related work's section and write a concise sentence instead.
As Professor McAuley (Amazon) states (in https://cseweb.ucsd.edu/~jmcauley/datasets.html): Please cite the following if you use the data: "Ups and downs: Modeling the visual evolution of fashion trends with one-class collaborative filtering", R. He, J. McAuley, WWW 2016 and "Image-based recommendations on styles and substitutes", J. McAuley, C. Targett, J. Shi, A. van den Hengel, SIGIR 2015
The references that you added seem to be incorrect: line 108: "Guo-Qiang et al. [23] ...", reference 23: "Gonzalezrodriguez M R ..." , line 111: "Wijayanti et al. [24] ...", reference 24: "Guo-Qiang Z..." . Please recheck all your references.
Please name all the papers' authors in the references' section. We use "et al." only in the paper's body, for brevity, however we must give the full references' information (authors, conference/journal, etc.) in the references' section.
I believe that algorithm is still hard to follow (even though the "for each" helps). Please use tabs (or extra spaces) to show which commands are in an if clause and/or for loop.
Author Response
Dear Editors and Reviewers,
We do appreciate your valuable suggestions on our paper entitled “A Recommendation Model based on Multi-emotion Similarity in the Social Networks (Manuscript ID: information-407433). After receiving the suggestions, we have made great efforts to carefully revise the manuscript according to the comments and suggestions of reviewers. The responses to reviewers are as follows:
Minor issues:
Point 1: Please, move the paragraph with the references 6-10 to the related work's section and write a concise sentence instead.
Response 1: Thank you for your suggestions. In the revised paper, we have carefully revised the introduction section and the related-work section, which are rewritten with the green color. We do hope that the revised paper can satisfy the strands of the Information Journal.
Point 2: As Professor McAuley (Amazon) states (in https://cseweb.ucsd.edu/~jmcauley/datasets.html): Please cite the following if you use the data: "Ups and downs: Modeling the visual evolution of fashion trends with one-class collaborative filtering", R. He, J. McAuley, WWW 2016 and "Image-based recommendations on styles and substitutes", J. McAuley, C. Targett, J. Shi, A. van den Hengel, SIGIR 2015
Response 2: Thank you for your advices. We have carefully read the papers given by the reviewers and compared those papers with the RM-MES scheme. The papers which have been added to the revised paper are shown in the following:
[1] He R , Mcauley J . Ups and Downs: Modeling the Visual Evolution of Fashion Trends with, One-Class Collaborative Filtering[C]// International Conference on World Wide Web. International World Wide Web Conferences Steering Committee, 2016.
[2] Mcauley J, Targett C, Shi Q, et al. Image-Based Recommendations on Styles and Substitutes[C]// International Acm Sigir Conference on Research & Development in Information Retrieval. 2015.
Point 3: The references that you added seem to be incorrect: line 108: "Guo-Qiang et al. [23] ...", reference 23: "Gonzalezrodriguez M R ..." line 111: "Wijayanti et al. [24] ...", reference 24: "Guo-Qiang Z..." . Please recheck all your references..
Response 3: Thank you for your advices, it is our negligence when writing this paper. We have carefully revised the whole references by word to avoid any mistakes. More rigorous attitude will be taken when revising the paper.
Point 4: Please name all the papers' authors in the references' section. We use "et al." only in the paper's body, for brevity, however we must give the full references' information (authors, conference/journal, etc.) in the references' section.
Response 4: Thank you for your advices. We have provided the full references' information in the revised paper.
Point 5: I believe that algorithm is still hard to follow (even though the "for each" helps). Please use tabs (or extra spaces) to show which commands are in an if clause and/or for loop.
Response 5: Thank you for your suggestions. We have revised this problem in the revised paper.
We hope that the revised paper can satisfy the requirements of the journal.
Sincerely yours,
Authors: Jun Long, Yulou Wang, Xinpan Yuan, Ting Li and Qunfeng Liu

Reviewer 2 Report
The paper might have been improved based on other reviewer's comments. However, my comments have been overlooked completely. The write-up is OK now but not perfect. Editing from a native English speaking expert is suggested again. The 'Reference Existing Store' is not detailed any further. Just 'three' new references have been added instead of five. The paragraph on 'Study limitations and future work' just serve the purpose of 'completing the formality'. There's no insight in it. Please improve it. In-text citation format is invalid. To cite a work, please follow this style:
Wang et al. [6] designed a combination model composed of the recommender and the similarity measure. Li et al. [7] proposed a trust-aware recommender system, which fully captures the influence of trust information and contextual information on ratings to improve the accuracy. Xiao-Yong et al. [8] proposed a latent social trust network model to improve the recommendation performance. Song et al. [9] researched how to achieve better recommendations of traditional recommendation models according to relationship information in social networks among customers and shops, and proposed a matrix decomposition framework based on integrating relationship information in social networks. Moreover, the probabilistic recommendation model is also widely used to solve the recommendation problem. Xu [10] proposed a probabilistic recommendation scheme with LDA topic model. These methods are generative because they conclude that some hidden processes produce usable data such as products, users, and ratings. To the shop owner, the effectiveness of a recommendation system depends on how accurately items are recommended to users.
Author Response
Dear Editors and Reviewers,
We do appreciate your valuable suggestions on our paper entitled “A Recommendation Model based on Multi-emotion Similarity in the Social Networks (Manuscript ID: information-407433). After receiving the suggestions, we have made great efforts to carefully revise the manuscript according to the comments and suggestions of reviewers. The responses to reviewers are as follows:
Point 1: The paper might have been improved based on other reviewer's comments. However, my comments have been overlooked completely. The write-up is OK now but not perfect. Editing from a native English speaking expert is suggested again.
Response 1: Thank you for your advices. We feel very sorry that the last revised paper didn’t meet the standards of your requirements. In the revised paper, we have carefully checked the words and grammars word by word to avoid any mistakes. What’s more, our manuscript has been checked and revised by a native English-speaking colleague. The sentences with the blue color are revised by a native English speaker and the sentences with the green color are revised by us. We do hope the revised paper can satisfy the minimum demands of the journal.
Point 2: The 'Reference Existing Store' is not detailed any further.
Response 2: Thank you for your suggestions, we have provided more explanations of the 'reference existing store' in the revised paper. The Revised portion are underlined in green.
Point 3: Just 'three' new references have been added instead of five.
Response 3: Thank you for your advices. We have added three new references in the revised paper. The papers which have been added to the revised paper are shown in the following:
[1] He R , Mcauley J . Ups and Downs: Modeling the Visual Evolution of Fashion Trends with, One-Class Collaborative Filtering[C]// International Conference on World Wide Web. International World Wide Web Conferences Steering Committee, 2016.
[2] Mcauley J, Targett C, Shi Q, et al. Image-Based Recommendations on Styles and Substitutes[C]// International Acm Sigir Conference on Research & Development in Information Retrieval. 2015.
[3] Musto, C., de Gemmis, M., Semeraro, G., Lops, P. (2017). A multi-criteria recommender system exploiting aspect-based sentiment analysis of users’ reviews. In Proceedings of the 11th ACM Conference on Recommender Systems (pp. 321–325).
Point 4: The paragraph on 'Study limitations and future work' just serve the purpose of 'completing the formality'. There's no insight in it. Please improve it.
Response 4: Thank you for your advices. In the revised paper, we have revised the section of study limitations and future directions. When the systems that have been running for some time, those items that have been recommended will be further recommended, and non-recommended items will suffer further reductions in their recommendation probabilities. Therefore, the recommendation system will become stuck in a closed state of over-maturation, and this state does not conform to the interests of the store. This is because the fact that the preferences of users cannot be fully explored. Therefore, we will further research how to recommend other goods with small probabilities to users in future works. This may breaks the weakness of a confined recommendation that exists in previous recommendation systems, and it enlarges the welfare of the system.
Point 5: In-text citation format is invalid. To cite a work, please follow this style:
Wang et al. [6] designed a combination model composed of the recommender and the similarity measure. Li et al. [7] proposed a trust-aware recommender system, which fully captures the influence of trust information and contextual information on ratings to improve the accuracy. Xiao-Yong et al. [8] proposed a latent social trust network model to improve the recommendation performance. Song et al. [9] researched how to achieve better recommendations of traditional recommendation models according to relationship information in social networks among customers and shops, and proposed a matrix decomposition framework based on integrating relationship information in social networks. Moreover, the probabilistic recommendation model is also widely used to solve the recommendation problem. Xu [10] proposed a probabilistic recommendation scheme with LDA topic model. These methods are generative because they conclude that some hidden processes produce usable data such as products, users, and ratings. To the shop owner, the effectiveness of a recommendation system depends on how accurately items are recommended to users.
Response 5: Thank you for your suggestions. We have revised this problem in the revised paper. The Revised portion are underlined in green.
We do hope that the revised paper can satisfy the strands of the Information Journal.
Sincerely yours,
Authors: Jun Long, Yulou Wang, Xinpan Yuan, Ting Li and Qunfeng Liu

Round 3
Reviewer 1 Report
Dear Authors,
Thank you for revising your paper.
My only comments are located in the references’ section.
Again, I can still see the “et al.” in many references (5, 6, 7, 10, 11, 12, 16, 17, 19, 21, 22, 27, 28).
Furthermore, the references are not written in the same way, eg.:
Ref 1. “…vol. 55, pp. 59–67, Sep. 2015.”
Ref 2. “…2015, 9.” (neither pages, nor volume/year)
Ref 12. “…2018:163-171.” (no “pp”, just the pages, plus “:” punctuation mark)
And this happens in all your references.
In some of them, you write “[J]” or “[C]” to indicate a Journal/Conference paper, while in some others you don’t.
Furthermore, in some references, do you use “//”?
Please follow the Journal’s guidelines, as far the references are concerned.
The reason I asked the inclusion of the two Mcauley’s references, in order to use the Amazon data in your research work (as Prof. Mcauley asks) is to place these two references, along with the “(http://jmcauley.ucsd.edu/data/amazon/)” when discussing the purchasing network metadata of Amazon product (line 356).
Apart from these issues, I am satisfied with the paper’s form and content.
I wish you a happy new year,
Reviewer2
Author Response
Dear Editors and Reviewers,
We do appreciate your valuable suggestions on our paper entitled “A Recommendation Model based on Multi-emotion Similarity in the Social Networks (Manuscript ID: information-407433). After receiving the suggestions, we have made great efforts to carefully revise the manuscript according to the comments and suggestions of reviewers. The responses to reviewers are as follows:
Point 1: My only comments are located in the references’ section.
Again, I can still see the “et al.” in many references (5, 6, 7, 10, 11, 12, 16, 17, 19, 21, 22, 27, 28).
Furthermore, the references are not written in the same way, eg.:
Ref 1. “…vol. 55, pp. 59–67, Sep. 2015.”
Ref 2. “…2015, 9.” (neither pages, nor volume/year)
Ref 12. “…2018:163-171.” (no “pp”, just the pages, plus “:” punctuation mark)
And this happens in all your references.
In some of them, you write “[J]” or “[C]” to indicate a Journal/Conference paper, while in some others you don’t.
Furthermore, in some references, do you use “//”?
Please follow the Journal’s guidelines, as far the references are concerned.
Response 1: Thank you for your suggestions. In the revised paper, we have carefully revised the whole references word by word to avoid any mistakes. We do hope that the revised paper can satisfy the strands of the Information Journal.
Point 2: The reason I asked the inclusion of the two Mcauley’s references, in order to use the Amazon data in your research work (as Prof. Mcauley asks) is to place these two references, along with the “(http://jmcauley.ucsd.edu/data/amazon/)” when discussing the purchasing network metadata of Amazon product (line 356).
Response 2: Thank you for your suggestions. We have revised this problem in the resubmitted paper.
We hope that the revised paper can satisfy the requirements of the journal.
Sincerely yours,
Authors: Jun Long, Yulou Wang, Xinpan Yuan, Ting Li and Qunfeng Liu

Reviewer 2 Report
The authors have made significant changes in the manuscript. Now, it seems me OK!
A review for English language usage of newly added sentences (colored as green) is suggested.
Author Response
Dear Editors and Reviewers,
We do appreciate your valuable suggestions on our paper entitled “A Recommendation Model based on Multi-emotion Similarity in the Social Networks (Manuscript ID: information-407433). After receiving the suggestions, we have made great efforts to carefully revise the manuscript according to the comments and suggestions of reviewers. The responses to reviewers are as follows:
Point 1: A review for English language usage of newly added sentences (colored as green) is suggested.
Response 1: Thank you for your advices. In the revised paper, we have carefully checked the newly added sentences word by word to avoid any mistakes. The Revised portion are underlined in gray. We do hope the revised paper can satisfy the minimum demands of the journal.
Sincerely yours,
Authors: Jun Long, Yulou Wang, Xinpan Yuan, Ting Li and Qunfeng Liu
